# Evaluation of Computer-Aided Detection (CAD) in Screening Automated Breast Ultrasound Based on Characteristics of CAD Marks and False-Positive Marks

**DOI:** 10.3390/diagnostics12030583

**Published:** 2022-02-24

**Authors:** Jeongmin Lee, Bong Joo Kang, Sung Hun Kim, Ga Eun Park

**Affiliations:** Department of Radiology, Seoul Saint Mary’s Hospital, College of Medicine, The Catholic University of Korea, Seoul 06591, Korea; jmlee328@gmail.com (J.L.); rad-ksh@catholic.ac.kr (S.H.K.); hoonhoony@naver.com (G.E.P.)

**Keywords:** computer-aided detection, automated breast ultrasound, breast

## Abstract

The present study evaluated the effectiveness of computer-aided detection (CAD) system in screening automated breast ultrasound (ABUS) and analyzed the characteristics of CAD marks and the causes of false-positive marks. A total of 846 women who underwent ABUS for screening from January 2017 to December 2017 were included. Commercial CAD was used in all ABUS examinations, and its diagnostic performance and efficacy in shortening the reading time (RT) were evaluated. In addition, we analyzed the characteristics of CAD marks and the causes of false-positive marks. A total of 1032 CAD marks were displayed based on the patient and 534 CAD marks on the lesion. Five cases of breast cancer were diagnosed. The sensitivity, specificity, PPV, and NPV of CAD were 60.0%, 59.0%, 0.9%, and 99.6% for 846 patients. In the case of a negative study, it was less time-consuming and easier to make a decision. Among 530 false-positive marks, 459 were identified clearly for pseudo-lesions; the most common cause was marginal shadowing, followed by Cooper’s ligament shadowing, peri-areolar shadowing, rib, and skin lesions. Even though CAD does not improve the performance of ABUS and a large number of false-positive marks were detected, the addition of CAD reduces RT, especially in the case of negative screening ultrasound.

## 1. Introduction

Mammographic screening has reduced the rate of breast cancer mortality [1]. Recent guidelines for screening of breast cancer recommend mammography starting at age 45 or 50 years [2,3]. Although the incidence of breast cancer in Asian women is still lower than in Western countries, morbidity and mortality continue to increase in Asian countries [4]. The peak age of breast cancer in Asian countries is 40–49 years, whereas in Western countries the peak is around 60 to 70 years [5]. Asian women tend to have breasts with higher density compared with Western women [6]. Further, dense breast is an independent risk factor for developing breast cancer [7].

Real-time B-mode ultrasonography has emerged as an alternative imaging technique for breast cancer screening [8]. Ultrasound elastography can quantify stiffness distribution of tissue lesions and complements conventional B-mode ultrasonography. The development of computer-aided diagnosis has improved the reliability of the system, whilst the inception of machine learning, such as deep learning, has further extended its power by facilitating automated segmentation and tumor classification [9].

Automated breast ultrasonography (ABUS) was proposed as a supplementary screening modality recently, for increased cancer detection combined with digital mammography (DM), especially in dense breasts [10,11,12]. In addition, ABUS has been proposed in the diagnostic setting in a few recent studies [13].

However, due to the large number of images in a single scan, the reading time (RT) of a full ABUS examination can be prolonged and cancers may be easily overlooked [14]. For this reason, computer-aided detection (CAD) software for ABUS has been developed to facilitate the radiological interpretation of ABUS examinations [15]. Few studies investigated the effect of commercially available CAD systems for ABUS on the RT and screening performance of breast radiologists [16]. However, before using the CAD system clinically, it is necessary to analyze the characteristics of CAD marks. It could be useful for radiologists to have knowledge about the characteristics of CAD marks and the causes of false-positive marks.

In this study, we evaluated the effectiveness of computer-aided detection (CAD) system in screening automated breast ultrasound (ABUS) through diagnostic performance and reading time (RT). We also investigated and analyzed the characteristics of CAD marks and the causes of false-positive marks, to distinguish between true and false marks.

## 2. Materials and Methods

This retrospective study was approved by the institutional review board (IRB) of our institution. The need for informed consent was waived by the ethics committee due to the retrospective design. All procedures involving human participants were in accordance with the ethical standards of IRB issued by our institution, and assessments were carried out in accordance with the tenets of the Declaration of Helsinki of 1975, and its revision in 2013.

### 2.1. ABUS Acquisitions

The ABUS examinations were performed with the ACUSON S2000 Automated Breast Volume Scanner system (Siemens, Erlangen, Germany). This ABUS system acquires 3D B-mode ultrasound volumes over an area of 15.4 × 16.8 × 6 cm^3^ volume data sets of the breast in one sweep using a mechanically driven linear array transducer (14L5). Adequate depth and focus can be obtained using predefined settings for different breast cup sizes. All ABUS examinations were performed by a single trained radiographer. To ensure coverage of the entire breast, three overlapping acquisitions including antero-posterior, medial, and lateral views were performed. The scan thickness was displayed at 1 mm intervals without overlap. A dedicated ABUS workstation was used to reconstruct the transverse slices into a 3D volume that can be read in a multiplanar hanging, with sagittal and coronal reconstructions.

### 2.2. CAD System

A prototype workstation was designed and developed specifically for high-throughput ABUS screening in this observer study (MeVis Medical Solutions, Bremen, Germany). In this prototype, each user action was logged with timestamps, which were subsequently used to estimate the time spent per case. The workstation was integrated with a commercially developed CAD software (QVCAD, Qview Medical Inc., Los Altos, CA, USA), which is designed to detect suspicious candidate regions in an ABUS volume highlighted with the so-called CAD marks (Figure 1).

In addition, the QVCAD software provides an “intelligent” minimum intensity projection (MinIP) of the breast tissue in a 3D ABUS volume that can be used for rapid navigation through ABUS scans for enhancement of the possible suspicious regions. The CAD-based MinIP integrated with a multiplanar hanging protocol for ABUS displays the conventional ABUS planes. By clicking on the dark spot, the 3D multiplanar hanging automatically snaps to the corresponding 3D location. The crosshair is focused on a breast lesion that is marked by the CAD software with a green circular marker. The same lesion is also enhanced and visualized as a dark spot in the MinIP. A screenshot of the CAD-aided reading environment is presented in Figure 1.

The number of CAD markers displayed per ABUS volume could be adjusted by changing the values of the false-positive rate (FPR) in the configuration setting of the CAD software. According to the manual from the manufacturer, FPR was defined as the total number of false-positive CAD markers in non-cancer volumes divided by the total number of non-cancer volumes. In this study, we set the FPR to 0.2 (i.e., 1 false-positive CAD marker in non-cancer volume per 5 non-cancer volumes), which was its default setting as in previous studies [16,17,18].

### 2.3. Study Design

The study included a total of 846 women aged 40–49 years who underwent ABUS screening from January 2017 to December 2017. The CAD (QVCADTM) system was used in all ABUS examinations and its diagnostic performance was evaluated retrospectively.

We evaluated glandular tissue component (GTC), which was classified as minimal (<25% of the fibroglandular tissue (FGT)), mild (25–49% of the FGT), moderate (50–74% of the FGT), or marked (≥75% of the FGT) in each woman based on bilateral breast images [19].

We analyzed whether CAD addition shortened the RT. The RT was determined by the expert breast radiologists based on their subjective perception in each of the following cases: (1) CAD with ABUS = ABUS only, (2) CAD with ABUS > ABUS only, (3) CAD with ABUS < ABUS only. We defined there is a difference when RT was shortened by more than 1 min.

Furthermore, we analyzed the characteristics of CAD marks including the size of the marked lesion, lesion type (mass or non-mass), tissue composition under ultrasound, and the causes of false-positive marks. The false-positive mark was defined as the mark located on the typical benign lesion or pseudo-lesions that require no additional studies following ABUS. The number of marks per patient and per lesion and the frequency of false-positive marks were also evaluated.

Two board-certified expert breast radiologists determined the characteristics of CAD marks based on consensus. In addition, the pseudo-lesions were also evaluated by two expert breast radiologists with consensus. The characteristics of pseudo-lesions were analyzed including the number, size, and location (right or left; antero-posterior, medial or lateral; upper, mid, or lower; inner, mid, or outer).

All women with suspicious lesions were recalled and US-guided 14G core-needle biopsy was performed. Patients who were not disease-positive were followed up in 2 years with radiologic examination using mammography or ultrasonography.

## 3. Results

A total of 846 women participated in the study, and the median age at enrollment was 44 years (mean age ± standard deviation = 43.9 ± 3.0 years). Based on ABUS screening, five breast cancers were diagnosed pathologically over a two-year follow-up (Figure 1). The sensitivity, specificity, PPV, NPV, and accuracy of CAD for cancer detection were 60.0%, 59.0%, 0.9%, 99.6% and 59.0%, respectively, for 846 patients, while those values for 1032 CAD marks were 60.0%, 48.3%, 0.6%, 99.6%, and 48.4%, respectively.

Based on the lesion type detected, the large mass lesions were more than the non-mass lesions (60 vs. 11). Based on tissue composition under ultrasound, the number of minimal-to-mild cases in GTC was higher than moderate-to-marked cases (668 vs. 178). The rate of CAD positivity in moderate-to-marked lesions was higher than in minimal-to-mild. Table 1 summarizes the screening performance of CAD for ABUS per patient and per lesion.

In the absence of the CAD mark, the readers determined that the reading time for CAD with ABUS was less than for ABUS only and easier to make a decision (Table 2). Table 2 summarizes the number and characteristics of CAD marks per patient.

Of 846 patients, 1032 CAD marks were marked in 534 lesions of 348 patients with a mean CAD mark per person of 0.8 (SD ± 1) (range 0–6) (Table 3). No CAD mark was detected in 498 patients (48.3%).

The characteristic CAD marks were determined by two reviewers by consensus as suspicious malignant lesions (0.8%, n = 4), benign lesions (13.3%. n = 71), and clear pseudo-lesions (86%, n = 459).

Among 530 false-positive marks, 459 marks were marked on the clear pseudolesions (Figure 2, Figure 3 and Figure 4); the most common cause was marginal shadowing (209, 39.1%), followed by Cooper’s ligament shadowing (143, 26.8%), peri-areolar shadowing (64, 12%), rib (37, 6.9 %), and skin lesions (6, 1.1%).

The false-positive marks on pseudo-lesions were frequently detected in the upper portion than in the mid-to-lower portion, and in the outer portion than in the mid-to-inner portion of breast (Table 4). There were more marks in the lateral view than in AP or medial views (Table 4).

## 4. Discussion

In this study, we evaluated the effectiveness of computer-aided detection (CAD) system in screening automated breast ultrasound (ABUS) through diagnostic performance and reading time (RT). A total of 846 patients displayed 1032 CAD marks and 534 CAD marks based on lesions. The sensitivity, specificity, PPV, NPV, and accuracy of CAD were 60.0%, 59.0%, 0.9%, 99.6% and 59.0% for 846 patients, respectively, while those of 1032 CAD marks were 60.0%, 48.3%, 0.6%, 99.6%, and 48.4%, respectively. The relatively higher NPV compared with other parameters indicates that the exam can be concluded with a negative study if no CAD mark is detected on ABUS. The presence of marks in multiple views did not suggest malignancy in this study. In the absence of the CAD mark, the readers determined that the reading time for CAD with ABUS was less than for ABUS only and easier to make a decision.

Several studies have reported that the performance of ABUS was comparable to that of hand-held ultrasound [20,21,22]. In addition, four prospective studies using ABUS demonstrated an increased cancer detection of 1.9–7.7 per 1000 examinations similar to hand-held ultrasound [10,11,14,23].

However, while the ABUS can yield standardized and structured images regardless of the experience of the operator, it takes much more time and effort to interpret the exams [24]. For this reason, the CAD system has been suggested as a supplementary method for interpreting ABUS results. However, the CAD system showed a high negative predictive value, and there were many false-positive CAD marks, which implied typical benign or pseudo-lesions that do not require further investigation. Usually, the false-positive imaging results can affect the recall rate of the screening modality. The recall rate varied from 8.8% in the J-START study to 10.7% in the American College of Radiology Imaging Network (ACRIN) study [25,26]. However, few studies reported the characteristics of the causes of false-positive marks.

In addition to the diagnostic performance of CAD on ABUS, the previous studies evaluated the RT of CAD on ABUS [27,28,29]. Yang et al. reported that using CAD in the concurrent-reading mode, all readers saved 32% (16 s per 50 s per volume) in RT with a higher area under the receiver operating characteristic curve values compared with non-CAD mode [28]. Jiang et al. reported that although not all studies were interpreted faster with the CAD system, on average the savings were approximately 1 min per case [29]. In our study, it was less time-consuming and easier to make a clinical decision, especially in the case of a negative study.

In this study, we investigated and analyzed the characteristics of CAD marks and the causes of false-positive marks, to distinguish between true and false marks. Among 530 false-positive marks, 459 were identified clearly for pseudo-lesions; the most common cause was marginal shadowing, followed by Cooper’s ligament shadowing, peri-areolar shadowing, rib, and skin lesions, all of which were easily distinguishable radiologically. The false marks for pseudo-lesions were detected more frequently in the upper rather than in the mid-to-lower portion and in the outer rather than in the mid-to-inner portion, probably because of bulkiness and flexibility of the upper and outer portion of the breast.

ABUS is a standardized examination with multiple advantages in both screening and diagnostic settings, including increased detection of breast cancer, improved workflow, and reduced examination time. However, ABUS has disadvantages and even some limitations. Disadvantages regarding image acquisition are the inability to assess the axilla, vascularization, and lesion elasticity. The limitations of interpretation include motion- or lesion-related artifacts due to poor positioning and the lack of contact [30]. In the review article about the pros and cons of ABUS by Ioana Boca et al., marginal shadowing and Cooper’s ligament shadowing were defined as artifacts due to insufficient compression [30]. Peri-areolar shadowing is defined as a nipple artifact [30]. Despite the promising detection rate with CAD software in breast cancer, radiologists should determine whether a CAD software-marked lesion is a true- or false-positive lesion, given its positive predictive value and high false-positive rate [17]. The knowledge of these artifacts improves the diagnostic performance of radiologists.

There are several limitations to this study. First, we used only image data obtained with equipment from a single vendor, with a small number of participants. In addition, this study was performed only in academic institutions by a limited number of users, board-certified expert breast radiologists, and does not represent varying clinical environments. Second, the absence of the numerical result of RT is the limitation of this study. The RT was determined by the expert breast radiologists based on their subjective perception. Finally, in our study, the expert radiologists’ decision was a gold standard for suspicious lesions or pseudo-lesions. However, a large number of marks await the radiologist’s rational judgment. Therefore, CAD users should be familiar with marks in various situations before using them, and the review summarizes the characteristics of CAD marks only without radiological evaluation. The knowledge of the characteristics of CAD marks and the causes of false-positive marks could improve the diagnostic performance of radiologists.

## 5. Conclusions

In conclusion, even though CAD addition does not improve the performance of screening ABUS and is associated with a large number of false-positive marks, CAD addition improves the negative predictive value and reduces RT, especially for negative screening ultrasound.

## Figures and Tables

**Figure 1 diagnostics-12-00583-f001:**
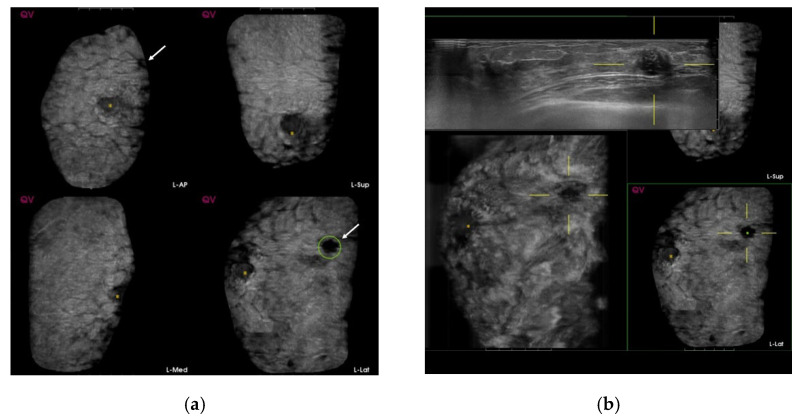
Screening automated breast ultrasound (ABUS) of a 44-year-old woman shows a true-positive mark. (**a**) Computer-aided detection (CAD)-based minimum intensity projection (MinIP) of an ABUS scan of the antero-posterior (AP), medial, and lateral sides of the left breast. There is one dark spot (arrows) with a green circle. (**b**) The lesion showing a dark spot with a green circle laterally on the left breast confirms invasive ductal carcinoma.

**Figure 2 diagnostics-12-00583-f002:**
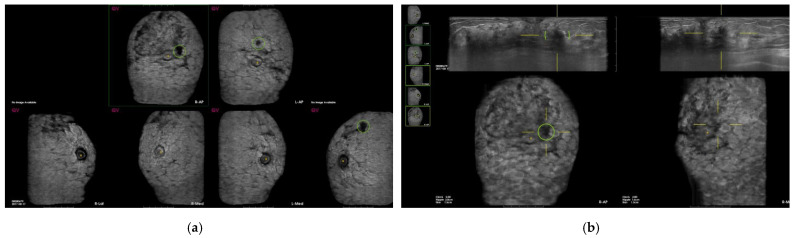
Screening automated breast ultrasound (ABUS) of a 45-year-old woman reveals false-positive marks due to shadowing. (**a**) CAD-based minimum intensity projection (MinIP) of an ABUS scan of the AP, medial, and lateral sides of both breasts. There are three dark spots with green circles. (**b**) The lesion showing a dark spot with a green circle on AP side of the right breast confirms the pseudolesion due to periareolar shadowing in the transverse scan. (**c**,**d**) The lesion showing a dark spot with a green circle on AP side of the left breast confirms the pseudolesion due to Cooper’s ligament shadowing in the transverse scan. The lesion showing a dark spot with a green circle laterally on the left breast confirms the pseudolesion due to marginal shadowing in the transverse scan.

**Figure 3 diagnostics-12-00583-f003:**
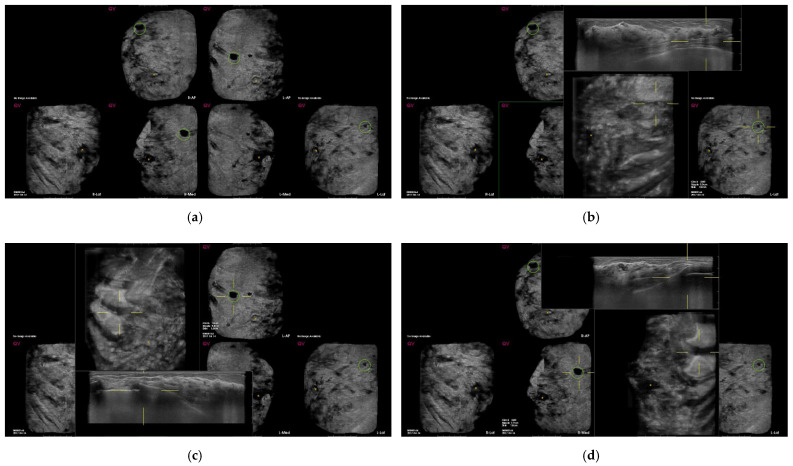
Screening automated breast ultrasound (ABUS) of a 42-year-old woman shows false-positive marks due to rib. (**a**) CAD-based minimum-intensity projection (MinIP) of an ABUS scan in the AP, medial, and lateral sides of both breasts. There are four dark spots with green circles. (**b**–**d**) The lesions showing dark spots with green circles in both AP and right medial sides of both breasts confirm pseudolesions due to ribs in the transverse scan.

**Figure 4 diagnostics-12-00583-f004:**
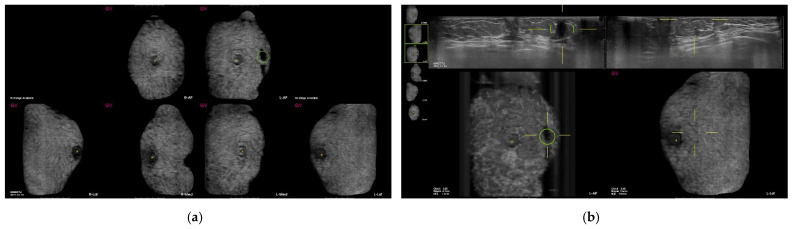
Screening automated breast ultrasound (ABUS) of a 48-year-old woman reveals false-positive marks due to skin lesions. (**a**) CAD-based minimum intensity projection (MinIP) of an ABUS scan in the AP, medial, and lateral sides of both breasts. There is a dark spot with a green circle. (**b**) The lesion showing a dark spot with a green circle on the AP side of the left breast confirms the pseudolesion due to a skin lesion in the transverse scan.

**Table 1 diagnostics-12-00583-t001:** Sensitivity (SEN), specificity (SPE), positive predictive value (PPV), negative predictive value (NPV), and accuracy per patient and per computer aided detection (CAD) mark.

Total (n = 846)	Benign	Malig	SEN	SPE	PPV	NPV	Accuracy	*p*-Value
**CAD**								
CAD (−)	496	2	60	59	0.9	99.6	59	0.407
CAD (+)	345	3						
Mark No. 1 ^#^	324	2						
Mark No. 2	20	-						
Mark No. 3	1	1						
**ABUS Category**								
<4	827	0	100	98.3	26.3	100	0.6	<0.0001
=>4	14	5						
**Lesion type**								
non-mass	10	1						
mass	56	4						
**Non-mass (n = 11)**								
CAD (−)	3	1	-	30	-	75	27.3	0.364
CAD (+)	7	0						
Mark No. 1	6	-						
Mark No. 2	1	-						
Mark No. 3	-	-						
**Mass (n = 60)**								
CAD (−)	27	1	75	48.2	9.4	96.4	50	0.616
CAD (+)	29	3						
Mark No. 1	23	2						
Mark No. 2	5	-						
Mark No. 3	1	1						
**Tissue Composition**								
1–2	665	3	40	79.1	1.1	99.6	78.8	0.284
3–4	176	2						
**Tissue Composition** **(1–2, n = 668)**								
CAD (−)	409	1	66.7	61.5	0.8	99.8	61.5	0.563
CAD (+)	256	2						
Mark No. 1	241	1						
Mark No. 2	14	-						
Mark No. 3	1	1						
**Tissue Composition** **(3–4, n = 178)**								
CAD (−)	87	1	50	49.4	1.1	98.9	49.4	1
CAD (+)	89	1						
Mark No. 1	83	1						
Mark No. 2	6	-						
Mark No. 3	-	-						
**Total (n = 1032)**								
CAD (−)	496	2	60	48.3	0.6	99.6	48.4	1
CAD (+)	531	3						
Mark Number 1 ^€^	220	3						
Mark Number 2	79	-						
Mark Number 3	36	-						
Mark Number 4	6	-						
Mark Number 5	3	-						
Mark Number 6	1	-						

^#^ Mark No. denotes the number of CAD marks per lesion. ^€^ Mark Number indicates the number of CAD marks per patient.

**Table 2 diagnostics-12-00583-t002:** Characteristics of number for computer-aided detection (CAD) marks per patient and reading time (RT).

	Mark No. ^#^				
	0	1	2, 3	Total (1,2,3)	*p*-Value *
**Size**					0.702
mean ± SD	12.2 ± 7.6	11.5 ± 6.3	18.9 ± 17.6	13 ± 9.9	
median(IQR)	10 (7, 14.5)	10 (7, 13)	15 (8, 20)	10 (7, 14)	
**Mass type**					0.743
non-mass	28 (87.5)	25 (80.7)	7 (87.5)	32 (82)	
mass	4 (12.5)	6 (19.4)	1 (12.5)	7 (18)	
**Tissue composition**					0.004
1, 2	410 (82.3)	242 (74.2)	16 (72.7)	258 (74.1)	
3, 4	88 (17.7)	84 (25.8)	6 (27.3)	90 (25.9)	
**Reading time**					<0.0001
CAD with ABUS = ABUS	16 (3.2)	39 (12.1)	10 (50)	49 (14.3)	
CAD with ABUS > ABUS	-	279 (86.4)	10 (50)	289 (84.3)	
CAD with ABUS < ABUS	482 (96.8)	5 (1.6)	-	5 (1.5)	

Values are expressed as numbers (percentages) for categorical variables and means (SD), median (IQR) others. * *p*-value was calculated between 0 with total (1,2,3) using Chi-square test, Fisher’s exact test, or *t*-test. ^#^ Mark No. indicates the number of CAD marks per lesion.

**Table 3 diagnostics-12-00583-t003:** Number and characteristics of computer-aided detection (CAD) mark.

Characteristics of All CAD Mark (n = 1032)		
**Mean and median No. of CAD marks per patient**		
mean ± SD	0.8 ± 1	
median (IQR)	1 (0, 1)	
**No. of CAD mark per patient**	**n**	**%**
0 (498)	498	48.3
1 (1 × 223)	223	21.6
2 (2 × 79)	158	15.3
3 (3 × 36)	108	10.5
4 (4 × 6)	24	2.3
5 (5 × 3)	15	1.5
6 (6 × 1)	6	0.6
**Characteristics of CAD marks per lesion (n = 534)**	**n**	**%**
**Suspicious**	4	0.8
**Benign**	71	13.3
Fat	35	6.6
Benign mass	19	3.6
Cyst	9	1.7
Fibrosis/heterogenous parenchyma	8	1.5
**False-positive marks for pseudolesions**	459	86
Marginal shadowing	209	39.1
Cooper’s ligament shadowing	143	26.8
Periareolar shadowing	64	12
Rib	37	6.9
Skin lesion	6	1.1

Values are expressed as numbers (percentages) for categorical variables and means (SD), median (IQR) others. Values are expressed as numbers (percentages) for categorical variables.

**Table 4 diagnostics-12-00583-t004:** Characteristics of false-positive marks associated with pseudolesions (n = 459).

	Mark No. ^#^				
	All	1	2, 3	*p*-Value	*p*-Value *
**Mark Location**				0.337	0.002
right	262 (57.1)	251 (56.7)	11 (68.8)		
left	197 (42.9)	192 (43.3)	5 (31.3)		
**Mark Location**				0.806	0.026
antero-posterior	142 (29.9)	131 (29.6)	11 (34.4)		
medial	147 (31)	137 (30.9)	10 (31.3)		
lateral	186 (39.2)	175 (39.5)	11 (34.4)		
**Mark Site**				0.674	<0.0001
upper	377 (82.1)	362 (81.7)	15 (93.8)		
mid	30 (6.5)	30 (6.8)	-		
lower	52 (11.3)	51 (11.5)	1 (6.3)		
**Mark Site**				0.572	<0.0001
inner	101 (22)	96 (21.7)	5 (31.3)		
mid	139 (30.3)	134 (30.3)	5 (31.3)		
outer	219 (47.7)	213 (48.1)	6 (37.5)		
**Tissue Composition**				0.843	<0.0001
1, 2	305 (66.5)	294 (66.4)	11 (68.8)		
3, 4	154 (33.6)	149 (33.6)	5 (31.3)		

Values represent numbers (percentages) for categorical variables. *p*-value was calculated between MarkNo1 with MarkNo2,3 using Chi-square test. * *p*-value was calculated only in a group using Chi-square test. ^#^ Mark No. indicates the number of CAD marks per lesion.

## Data Availability

All data generated and analyzed during this study are included in this published article. Raw data supporting the findings of this study are available from the corresponding author on request.

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
