# Peer review of "Evaluation of Computer-Aided Detection (CAD) in Screening Automated Breast Ultrasound Based on Characteristics of CAD Marks and False-Positive Marks"

_diagnostics, 2022, doi:10.3390/diagnostics12030583_

Round 1
Reviewer 1 Report
In this paper, the authors study the impact of using automatic breast ultrasound and CAD systems for malign marks on breast imaging. The rigorous and thorough analysis of CAD systems is crucial in order to understand the real potential of these tools and their diagnosis value. It is also very important to know the reasons for making mistakes when labeling marks so the community can work on that, specially nowadays, when most CAD systems use deep learning-based methods, which, despite showing great performance, they tend so suffer from a black-box effect. This definitely affect the trust of the community on this technology.
Overall, the paper is well written and in can be easily followed. However, there are some points that need to be clarified. Please, see more comments below.
- In my experience, a proper set-up can make a big difference in terms of the diagnostic capacity and image quality. Could the authors explain a bit more on that. “Using predefined settings for different breast cup sizes” doesn’t say much.
- “Each user action was logged with time stamps” à I guess it is possible, based on that information, to also analyze the distribution of the different actions. On which type of action the radiologist spend more time, etc.
- If you use and explain the use of MinIP as short form for “minimum intensity projection” once, you don’t have to keep using both together again. Check line 79 and 81.
- “In this study, we selected the default setting of a single false-positive CAD mark in each ABUS volume” -> This configuration option is very important, since it is likely the sensitivity is directly affected. By using the “single false-positive” option will likely reduce the number of marks in general, thus conditioning the average exploration time of the marks proposed. This actually means that the entire study is conditioned by this very value, and ideally, it should be done for different values of this option. Please, comment on this since it seems to be a critical configuration option.
- Please, include the demographic statistics of the patients in the study (average age, median, std, etc).
- Regarding the high NPV value, how were the False Negative cases calculated when computing the NPV?
- When comparing the performance for patients (846) vs marks (1032), most of the metrics seems to be in the same range, but the PPV, going from 0.9 for patients to 0.6 for marks. I guess that due to the low number of false positive patients, meaning some of the marks are not FP. Please, comment on that.
- In Table 2, what dies the reading time actually indicate? The number of cases on each category (=,<,>)? Please, indicate also the time distribution (in seconds). It is interesting to categorize the comparison into three groups, but it is also important to know how far is one alternative from the other. CAD+ABUS > ABUS can be 1second off. Plus, what is the criteria to stablish CAD+ABUS = ABUS? What is the tolerance? Please, explain and clarify.
Even though the study was limited to one single US vendor and one single CAD system, do the authors have further information regarding the origin of false positive cases in other studies or using different systems?
Author Response
Response to Reviewer 1 Comments
In this paper, the authors study the impact of using automatic breast ultrasound and CAD systems for malign marks on breast imaging. The rigorous and thorough analysis of CAD systems is crucial in order to understand the real potential of these tools and their diagnosis value. It is also very important to know the reasons for making mistakes when labeling marks so the community can work on that, specially nowadays, when most CAD systems use deep learning-based methods, which, despite showing great performance, they tend so suffer from a black-box effect. This definitely affect the trust of the community on this technology.
Overall, the paper is well written and in can be easily followed. However, there are some points that need to be clarified. Please, see more comments below.
In my experience, a proper set-up can make a big difference in terms of the diagnostic capacity and image quality. Could the authors explain a bit more on that. “Using predefined settings for different breast cup sizes” doesn’t say much.
-> Response: We agree with you. Nowadays, we know that the appropriate operating point of this technique is almost fixed through several studies, and we used it.
However, in Korea (for Asian women), the range of cup size is not wide and not important for this examination (ABUS and CAD), and the usual setting is proper for almost women. So, we did not have any difficulty about set-up during this study. We paid more attention to the training of the radiologist rather than the set-up, and made an effort to be familiar with the examination and to perform the examination consistently.
“Each user action was logged with time stamps” à I guess it is possible, based on that information, to also analyze the distribution of the different actions. On which type of action the radiologist spend more time, etc.
-> Response: We defined there is a difference when RT was shortened by more than 1 minute. If two people chose different action, then we choose the one that fits this definition (more than 1 minute difference). There was rare different action in this study. We added it in the method section.
If you use and explain the use of MinIP as short form for “minimum intensity projection” once, you don’t have to keep using both together again. Check line 79 and 81.
-> Response: We corrected line 79 and 81.
“In this study, we selected the default setting of a single false-positive CAD mark in each ABUS volume” -> This configuration option is very important, since it is likely the sensitivity is directly affected. By using the “single false-positive” option will likely reduce the number of marks in general, thus conditioning the average exploration time of the marks proposed. This actually means that the entire study is conditioned by this very value, and ideally, it should be done for different values of this option. Please, comment on this since it seems to be a critical configuration option.
-> Response: We agree with you. Nowadays, we know that the appropriate operating point of this technique is almost fixed through several studies, and we used it. We added more comment and references (16-18).
Please, include the demographic statistics of the patients in the study (average age, median, std, etc).
-> Response: We added the demographic statistics of the patients.
Regarding the high NPV value, how were the False Negative cases calculated when computing the NPV?
-> Response:. Patients who were not disease-positive were followed up in 2 years with radiologic examination using mammography or ultrasonography (HHUS or ABUS). We clarified it.
When comparing the performance for patients (846) vs marks (1032), most of the metrics seems to be in the same range, but the PPV, going from 0.9 for patients to 0.6 for marks. I guess that due to the low number of false positive patients, meaning some of the marks are not FP. Please, comment on that.
-> Response: As you can see the table 1, when comparing the performance for patients (846) vs marks (1032), cad (+) were 345/846 vs 531/1032. In one patient, several number of false positive marker are demonstrated. I guess that it’s the reason for PPV, going from 0.9 for patients to 0.6 for marks.
In Table 2, what dies the reading time actually indicate? The number of cases on each category (=,<,>)? Please, indicate also the time distribution (in seconds). It is interesting to categorize the comparison into three groups, but it is also important to know how far is one alternative from the other. CAD+ABUS > ABUS can be 1second off. Plus, what is the criteria to stablish CAD+ABUS = ABUS? What is the tolerance? Please, explain and clarify.
-> Response:. We agreed with you. The absence of the numerical result of RT is the limitation of this study. The RT was determined by the expert breast radiologists based on their subjective perception. We added about it in the limitation section.
Even though the study was limited to one single US vendor and one single CAD system, do the authors have further information regarding the origin of false positive cases in other studies or using different systems?
-> Response:.I don’t find different system or multicenter study. There was a few reports retrospective observer performance study. We included those references.
Reviewer 2 Report
The introduction section is too short without the background of ultrasound imaging. It should be including a brief overview of ultrasound imaging techniques including B-mode imaging, 3D ultrasound and ultrasound elastography for the breast. Please refer to and include the following article.
Mao, Y.-J.; Lim, H.-J.; Ni, M.; Yan, W.-H.; Wong, D.W.-C.; Cheung, J.C.-W. Breast Tumour Classification Using Ultrasound Elastography with Machine Learning: A Systematic Scoping Review. Cancers 2022, 14, 367. https://doi.org/10.3390/cancers14020367
Figure 1 should be split into 2 and enhance the markers.
Table 3 is not in format in a good format. The percentage value should be filled in another column.
The key finding of this study and importance of this study should be discussed at the beginning of the discussion section.
RT is one of the key components of the study. The numerical result is absent in the abstract and discusses in the earlier discussion section
Author Response
Response to Reviewer 2 Comments
The introduction section is too short without the background of ultrasound imaging. It should be including a brief overview of ultrasound imaging techniques including B-mode imaging, 3D ultrasound and ultrasound elastography for the breast. Please refer to and include the following article.
Mao, Y.-J.; Lim, H.-J.; Ni, M.; Yan, W.-H.; Wong, D.W.-C.; Cheung, J.C.-W. Breast Tumour Classification Using Ultrasound Elastography with Machine Learning: A Systematic Scoping Review. Cancers 2022, 14, 367. https://doi.org/10.3390/cancers14020367
-> Response: We added a brief overview of screening and ultrasound imaging techniques, as you recommended
Figure 1 should be split into 2 and enhance the markers.
-> Response: We added arrows in the figure 1. Unfortunately, we don’t have additional valid image stored, so, we corrected the figure 1 as we can.
Table 3 is not in format in a good format. The percentage value should be filled in another column.
-> Response: We corrected the Table 3 as you recommended.
The key finding of this study and importance of this study should be discussed at the beginning of the discussion section.
-> Response: We added them at the beginning of the discussion section.
RT is one of the key components of the study. The numerical result is absent in the abstract and discusses in the earlier discussion section
-> Response: We agreed with you. The absence of the numerical result of RT is the limitation of this study. We added about it in the earlier discussion section and limitation.
Round 2
Reviewer 2 Report
I appreciate the efforts.